# LEARNING TWITTER USER SENTIMENTS ON CLIMATE CHANGE WITH LIMITED LABELED DATA

## ABSTRACT

While it is well-documented that climate change accepters and deniers have become increasingly polarized in the United States over time (McCright & Dunlap, 2011), there has been no large-scale examination of whether these individuals are prone to changing their opinions as a result of natural external occurrences. On the sub-population of Twitter users, we examine whether climate change sentiment changes in response to five separate natural disasters occurring in the U.S. in 2018. We begin by showing that tweets can be classified with over 75% accuracy as either accepting or denying climate change when using our methodology to compensate for limited labelled data; results are robust across several machine learning models and yield geographic-level results in line with prior research (Howe et al., 2015). We then apply RNNs to conduct a cohort-level analysis showing that the 2018 hurricanes yielded a statistically significant increase in average tweet sentiment affirming climate change. However, this effect does not hold for the 2018 blizzard and wildfires studied, implying that Twitter users' opinions on climate change are fairly ingrained on this subset of natural disasters.

## 1 BACKGROUND

Much prior work has been done at the intersection of climate change and Twitter, such as tracking climate change sentiment over time (An et al., 2014), finding correlations between Twitter climate change sentiment and seasonal effects (Baylis, 2015), and clustering Twitter users based on climate mentalities using network analysis (Swain, 2016). Throughout, Twitter has been accepted as a powerful tool given the magnitude and reach of samples unattainable from standard surveys. However, the aforementioned studies are not scalable with regards to training data, do not use more recent sentiment analysis tools (such as neural nets), and do not consider unbiased comparisons pre- and post- various climate events (which would allow for a more concrete evaluation of shocks to climate change sentiment). This paper aims to address these three concerns as follows.

First, we show that machine learning models formed using our labeling technique can accurately predict tweet sentiment (see Section 3.1). We introduce a novel method to intuit binary sentiments of large numbers of tweets for training purposes. Second, we quantify unbiased outcomes from these predicted sentiments (see Section 3.2). We do this by comparing sentiments within the same cohort of Twitter users tweeting both before and after specific natural disasters; this removes bias from over-weighting Twitter users who are only compelled to compose tweets after a disaster.

## 2 DATA

We henceforth refer to a tweet affirming climate change as a "positive" sample (labeled as 1 in the data), and a tweet denying climate change as a "negative" sample (labeled as -1 in the data). All data were downloaded from Twitter in two separate batches using the "twint" scraping tool (Zacharias, 2018) to sample historical tweets for several different search terms; queries always included either "climate change" or "global warming", and further included disaster-specific search terms (e.g., "bomb cyclone," "blizzard," "snowstorm," etc.). We refer to the first data batch as "influential" tweets, and the second data batch as "event-related" tweets.

The first data batch consists of tweets relevant to blizzards, hurricanes, and wildfires, under the constraint that they are tweeted by "influential" tweeters, who we define as individuals certain to have

a classifiable sentiment regarding the topic at hand. For example, we assume that any tweet composed by Al Gore regarding climate change is a positive sample, whereas any tweet from conspiracy account @ClimateHiJinx is a negative sample. The assumption we make in ensuing methods (confirmed as reasonable in Section 3.1) is that influential tweeters can be used to label tweets in bulk in the absence of manually-labeled tweets. Here, we enforce binary labels for all tweets composed by each of the 133 influential tweeters that we identified on Twitter (87 of whom accept climate change), yielding a total of 16,360 influential tweets.

The second data batch consists of event-related tweets for five natural disasters occurring in the U.S. in 2018. These are: the East Coast Bomb Cyclone (Jan. 2 - 6); the Mendocino, California wildfires (Jul. 27 - Sept. 18); Hurricane Florence (Aug. 31 - Sept. 19); Hurricane Michael (Oct. 7 - 16); and the California Camp Fires (Nov. 8 - 25). For each disaster, we scraped tweets starting from two weeks prior to the beginning of the event, and continuing through two weeks after the end of the event. Summary statistics on the downloaded event-specific tweets are provided in Table 1. Note that the number of tweets occurring prior to the two 2018 sets of California fires are relatively small. This is because the magnitude of these wildfires were relatively unpredictable, whereas blizzards and hurricanes are often forecast weeks in advance alongside public warnings. The first (influential tweet data) and second (event-related tweet data) batches are de-duplicated to be mutually exclusive. In Section 3.1, we perform geographic analysis on the event-related tweets from which we can scrape self-reported user city from Twitter user profile header cards; overall this includes 840 pre-event and 5,984 post-event tweets.

To create a model for predicting sentiments of event-related tweets, we divide the first data batch of influential tweets into training and validation datasets with a 90%/10% split. The training set contains 49.2% positive samples, and the validation set contains 49.0% positive samples. We form our test set by manually labelling a subset of 500 tweets from the the event-related tweets (randomly chosen across all five natural disasters), of which 50.0% are positive samples.

Table 1: Tweets collected for each U.S. 2018 natural disaster

| Natural Disaster | Total Tweets | # Pre-Event | # Post-Event | # Users Tweeting Pre- and Post-Event | # Same-User Pre-Event Tweets | # Same-User Post-Event Tweets |
|---|---|---|---|---|---|---|
| Bomb Cyclone | 15,080 | 2,632 | 12,488 | 456 | 1,115 | 3,439 |
| Mendocino Wildfire | 3,056 | 173 | 2,883 | 36 | 49 | 103 |
| Hurricane Florence | 6,597 | 814 | 5,783 | 131 | 224 | 527 |
| Hurricane Michael | 13,606 | 1,965 | 11,641 | 301 | 917 | 1,426 |
| Camp Fire | 6,774 | 55 | 6,719 | 14 | 19 | 43 |

## 3 METHODS

### 3.1 PROOF OF CONCEPT FOR LABELLING METHODOLOGY

Our first goal is to train a sentiment analysis model (on training and validation datasets) in order to perform classification inference on event-based tweets. We experimented with different feature extraction methods and classification models. Feature extractions examined include Tokenizer, Unigram, Bigram, 5-char-gram, and td-idf methods. Models include both neural nets (e.g. RNNs, CNNs) and standard machine learning tools (e.g. Naive Bayes with Laplace Smoothing, k-clustering, SVM with linear kernel). Model accuracies are reported in Table 2.

The RNN pre-trained using GloVe word embeddings (Pennington et al., 2014) achieved the higest test accuracy. We pass tokenized features into the embedding layer, followed by an LSTM (Hochreiter & Schmidhuber, 1997) with dropout and ReLU activation, and a dense layer with sigmoid activation. We apply an Adam optimizer on the binary crossentropy loss. Implementing this simple, one-layer LSTM allows us to surpass the other traditional machine learning classification methods. Note the 13-point spread between validation and test accuracies achieved. Ideally, the training, validation, and test datasets have the same underlying distribution of tweet sentiments; the assumption made with our labelling technique is that the influential accounts chosen are representative of all Twitter accounts.

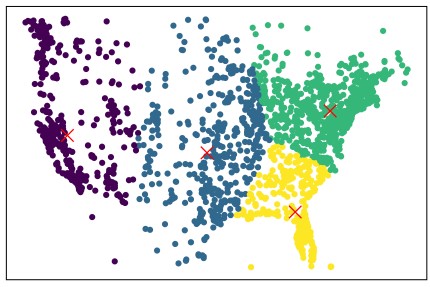

Figure 1: Four-clustering on sentiment, latitude, and longitude

Table 2: Selected binary sentiment analysis accuracies

| Method | Validation Accuracy | Test Accuracy |
|---|---|---|
| Tokenizer RNN | 88.7% | 75.4% |
| Unigram SVM | 86.6% | 74.6% |
| 5-char-gram Naive Bayes | 87.2% | 73.5% |
| Unigram Naive Bayes | 88.6% | 71.4% |
| td-idf Naive Bayes | 87.8% | 70.8% |
| Unigram k-means | 60.4% | 58.0% |

Critically, when choosing the influential Twitter users who believe in climate change, we highlighted primarily politicians or news sources (i.e., verifiably affirming or denying climate change); these tweets rarely make spelling errors or use sarcasm. Due to this skew, the model yields a high rate of false negatives. It is likely that we could lessen the gap between validation and test accuracies by finding more "real" Twitter users who are climate change believers, e.g. by using the methodology found in Swain (2016).

### 3.2 OUTCOME ANALYSIS

Our second goal is to compare the mean values of users' binary sentiments both pre- and post- each natural disaster event. Applying our highest-performing RNN to event-related tweets yields the following breakdown of positive tweets: Bomb Cyclone (34.7%), Mendocino Wildfire (80.4%), Hurricane Florence (57.2%), Hurricane Michael (57.6%), and Camp Fire (70.1%). As sanity checks, we examine the predicted sentiments on a subset with geographic user information and compare results to the prior literature.

In Figure 1, we map 4-clustering results on three dimensions: predicted sentiments, latitude, and longitude. The clusters correspond to four major regions of the U.S.: the Northeast (green), Southeast (yellow), Midwest (blue), and West Coast (purple); centroids are designated by crosses. Average sentiments within each cluster confirm prior knowledge (Howe et al., 2015): the Southeast and Midwest have lower average sentiments (-0.17 and -0.09, respectively) than the West Coast and Northeast (0.22 and 0.09, respectively). In Figure 2, we plot predicted sentiment averaged by U.S. city of event-related tweeters. The majority of positive tweets emanate from traditionally liberal hubs (e.g. San Francisco, Los Angeles, Austin), while most negative tweets come from the Philadelphia metropolitan area. These regions aside, rural areas tended to see more negative sentiment tweeters post-event, whereas urban regions saw more positive sentiment tweeters; however, overall average climate change sentiment pre- and post-event was relatively stable geographically. This map further confirms findings that coastal cities tend to be more aware of climate change (Milfont et al., 2014).

From these mapping exercises, we claim that our "influential tweet" labelling is reasonable. We now discuss our final method on outcomes: comparing average Twitter sentiment pre-event to post-event. In Figure 3, we display these metrics in two ways: first, as an overall average of tweet binary sentiment, and second, as a within-cohort average of tweet sentiment for the subset of tweets by users who tweeted both before and after the event (hence minimizing awareness bias). We use Student's t-test to calculate the significance of mean sentiment differences pre- and post-event (see Section 4). Note that we perform these mean comparisons on all event-related data, since the low number of geo-tagged samples would produce an underpowered study.

## 4 RESULTS & DISCUSSION

In Figure 3, we see that overall sentiment averages rarely show movement post-event: that is, only Hurricane Florence shows a significant difference in average tweet sentiment pre- and post-event at the 1% level, corresponding to a 0.12 point decrease in positive climate change sentiment. However, controlling for the same group of users tells a different story: both Hurricane Florence and Hurricane

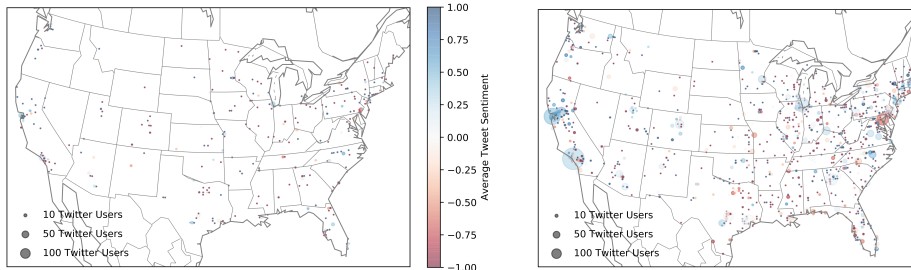

Figure 2: Pre-event (left) and post-event (right) average climate sentiment aggregated over five U.S. natural disasters in 2018

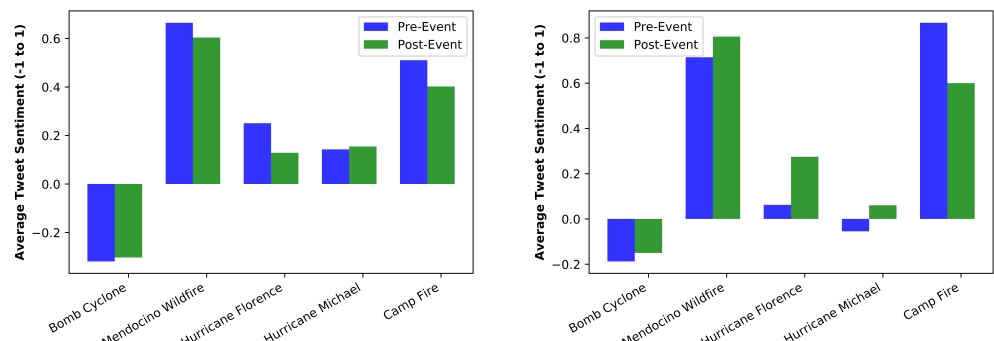

Figure 3: Comparisons of overall (left) and within-cohort (right) average sentiments for tweets occurring two weeks before or after U.S. natural disasters occurring in 2018

Michael have significant tweet sentiment average differences pre- and post-event at the 1% level. Within-cohort, Hurricane Florence sees an increase in positive climate change sentiment by 0.21 points, which is contrary to the overall average change (the latter being likely biased since an influx of climate change deniers are likely to tweet about hurricanes only after the event). Hurricane Michael sees an increase in average tweet sentiment of 0.11 points, which reverses the direction of tweets from mostly negative pre-event to mostly positive post-event. Likely due to similar bias reasons, the Mendocino wildfires in California see a 0.06 point decrease in overall sentiment post-event, but a 0.09 point increase in within-cohort sentiment. Methodologically, we assert that overall averages are not robust results to use in sentiment analyses.

We now comment on the two events yielding similar results between overall and within-cohort comparisons. Most tweets regarding the Bomb Cyclone have negative sentiment, though sentiment increases by 0.02 and 0.04 points post-event for overall and within-cohort averages, respectively. Meanwhile, the California Camp Fires yield a 0.11 and 0.27 point sentiment decline in overall and within-cohort averages, respectively. This large difference in sentiment change can be attributed to two factors: first, the number of tweets made regarding wildfires prior to the (usually unexpected) event is quite low, so within-cohort users tend to have more polarized climate change beliefs. Second, the root cause of the Camp Fires was quickly linked to PG&E, bolstering claims that climate change had nothing to do with the rapid spread of fire; hence within-cohort users were less vocally positive regarding climate change post-event.

There are several caveats in our work: first, tweet sentiment is rarely binary (this work could be extended to a multinomial or continuous model). Second, our results are constrained to Twitter users, who are known to be more negative than the general U.S. population (Caumont et al., 2013). Third, we do not take into account the aggregate effects of continued natural disasters over time. Going forward, there is clear demand in discovering whether social networks can indicate environmental metrics in a "nowcasting" fashion. As climate change becomes more extreme, it remains to be seen what degree of predictive power exists in our current model regarding climate change sentiments with regards to natural disasters.

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
