# OpenReview forum: "Learning Twitter User Sentiments on Climate Change with Limited Labeled Data"
_ICLR.cc/2019/Workshop/LLD — Submitted to LLD 2019_

### Official Review · AnonReviewer1 · 2019-04-05
**Interesting twitter analysis of reactions to climate change**

**Rating:** 2
**Confidence:** 2

**Review:**

This paper describes the collection and analysis of tweets expressing denial or acceptance of climate change. Training and development data for supervised training of a binary tweet classifier are collected from influential tweeters with known opinions towards climate change. The trained classifier is then used to classify a larger collection of tweets around natural disasters in the US and its predictions are analyzed, with respect to geolocation and change over time.

Pros:
- Interesting insights into reaction on twitter towards climate change.
- Comparison of a diverse set of classic binary classification algorithms.
- Good visualizations and discussion of outcomes, including significance tests.

Cons:
- Data: Is the data made publicly available? How were the influencers chosen?
- Analysis: It's not clear whether the classifier is reliable enough to draw these conclusions. Examples from the classifications would allow a rough inspection of the quality. If a different classifier would have been chosen, would the conclusions from the analysis still hold?
- The trick for dealing with unlimited data is not in the focus of this paper and also not evaluated against any other method.

In summary, I find the analysis of the classified tweets interesting and neat, but my background knowledge on twitter analysis for climate change is too limited to assess whether its novelty. However, possible inaccuracies or biases of the deployed classifier are not discussed and the aspect of learning with limited data is not in the focus of the paper, which makes me question the relevance for this workshop.

---

### Official Review · AnonReviewer2 · 2019-04-07
**Good paper, perhaps a better fit for a different workshop**

**Rating:** 2
**Confidence:** 2

**Review:**

The paper investigates sentiment with respect to climate change and how this changes in tweets after specific natural disasters.
Tweets from hand-picked influencers are collected, such that these tweets can be assumed to have a specific label and are then used for training a climate change sentiment classifier.
Several well-established models are compared, with the RNN model achieving the best results.
The classifier is then applied on a wider selection of tweets to measure the change in sentiment before and after specific natural disasters.
The results show that the sentiment moves slightly towards the positive after some hurricanes, but is largely not affected by the natural disasters.

The paper is an interesting pilot work and makes some useful contributions, but not necessarily in the target area of the LLD 2019 Workshop.
The workshop focus is on representation learning with limited data, whereas the paper does not address representation learning and the novelty of the limited data part is minimal (selecting specific users whose tweets are assumed to have the same label).

There are definitely interesting contributions here in the area of social sciences and social media research. The analysis of sentiment timed with specific events was interesting to see.
The paper also identifies specific shortcomings of the current work, which is useful as a pilot experiment and can drive future work.

It seems unlikely that out of 500 randomly sampled tweets exactly 50.0% were positive and negative. Was there actually a more imbalanced distribution that was specifically corrected?

The images in Figure 2 are too small to be interpretable.

---

### Decision · Program_Chairs · 2019-04-08
**Acceptance Decision**

Reject